# Species Distribution of Candidemia and Their Susceptibility in a Single Japanese University Hospital: Prior Micafungin Use Affects the Appearance of *Candida parapsilosis* and Elevation of Micafungin MICs in Non-*parapsilosis Candida* Species

**DOI:** 10.3390/jof7080596

**Published:** 2021-07-23

**Authors:** Yasutaka Sakamoto, Kazuhiro Kawabe, Tomoyo Suzuki, Kayoko Sano, Kazuo Ide, Tetsuta Nishigaki, Yuki Enoki, Kazuaki Taguchi, Hirofumi Koike, Hideaki Kato, Yukiko Sahashi, Kazuaki Matsumoto

**Affiliations:** 1Division of Pharmacodynamics, Keio University Faculty of Pharmacy, Tokyo 105-8512, Japan; yasutaka@yokohama-cu.ac.jp (Y.S.); taguchi-kz@pha.keio.ac.jp (K.T.); matsumoto-kz@pha.keio.ac.jp (K.M.); 2Department of Pharmacy, Yokohama City University Hospital, Yokohama 236-0004, Japan; k_kawabe@yokohama-cu.ac.jp (K.K.); koyama_t@yokohama-cu.ac.jp (T.S.); kazuo_i@yokohama-cu.ac.jp (K.I.); ntetsuta@yokohama-cu.ac.jp (T.N.); hi00_k@yokohama-cu.ac.jp (H.K.); ysahashi@yokohama-cu.ac.jp (Y.S.); 3Department of Clinical Laboratory, Yokohama City University Hospital, Yokohama 236-0004, Japan; ksano@yokohama-cu.ac.jp; 4Infection Prevention and Control Department, Yokohama City University Hospital, Yokohama 236-0004, Japan; ekato@yokohama-cu.ac.jp

**Keywords:** candidemia, risk factor, micafungin, *C. parapsilosis*, non-*albicans Candida* species, minimum inhibitory concentrations, low susceptibility, antifungal prescription, prior antifungal use

## Abstract

Introduction: Micafungin is a recommended echinocandin antifungal agent for candidemia treatment and prophylaxis. However, overuse of echinocandin antifungals may cause resistance. There is currently no information available regarding the low susceptibility associated with using micafungin. This study investigated the effect of micafungin use on changes in the detected *Candida* species and low susceptibility. Methods: We conducted a retrospective survey and included records of *Candida* spp. detected in blood cultures from January 2010 to December 2018 in our hospital. Survey items included clinical outcomes at 30 days after positive cultures, patient characteristics, and drug prescription status. Patient background information included gender, previous hospitalization, stay in the intensive care unit, comorbidities, and history of surgery (within 90 days before candidemia onset) and drug exposure. Species detected and their minimum inhibitory concentrations (MICs) and amount of antifungal prescriptions by department were investigated. Risk factors for detecting *C. parapsilosis* and for low susceptibility to micafungin were evaluated using multivariate analysis. Results: A total of 153 *Candida* clinical blood isolates were collected and *C. albicans* was the most prevalent species, followed by *C. parapsilosis* and *C. glabrata*. In the analysis by department, antifungal use and non-*albicans Candida* species were most frequently detected in the hematology department. Multivariate analysis showed that prior micafungin use increased the risk of *C. parapsilosis* (odds ratio (OR) 4.22; 95% confidence interval (CI) 1.39–12.79; *p* = 0.011). MIC_90_ of micafungin on *C. glabrata* and *C. parapsilosis* was 1.0 μg/mL. Prior micafungin use was clarified as a risk factor resulting in MIC > 0.06 μg/mL for micafungin in non-*parapsilosis Candida* species (OR 13.2; 95% CI 3.23–54.2; *p* < 0.01). Conclusion: Prior micafungin use increased the risk of *C. parapsilosis* and the MIC > 0.06 μg/mL of micafungin in non-*parapsilosis Candida* species. Since there are only a few antifungal options, further antifungal stewardship considering azole antifungal agents use is required.

## 1. Introduction

Candidemia is one cause of nosocomial bloodstream infections [1]. The mortality rate from candidemia was 39.3% in Japan [2], 38–39.2% in the United States [3,4], and 19–38.8% in Europe [5,6] with a higher mortality rate than other nosocomial bloodstream infections in the world. Therefore, adequately treating candidemia is critical for improving patient prognosis [7,8].

Micafungin is an echinocandin antifungal agent used for the treatment and prevention of candidemia. The Infectious Diseases Society of America guidelines recommend micafungin, caspofungin, and anidulafungin as the initial therapy for non-neutropenic and neutropenic patients [9]. The European Society of Clinical Microbiology and Infectious Diseases guidelines recommend micafungin for the initial treatment of candidemia in non-neutropenic and neutropenic patients [10] and also for prophylaxis against allogeneic hematopoietic stem cell recipients [11]. In the Japanese Domestic Guidelines for Management of Deep-seated Mycosis 2014, micafungin is recommended as an initial treatment for candidemia in severe non-neutropenic and neutropenic patients [12,13]. Micafungin is also recommended for prophylaxis after allogeneic hematopoietic stem cell transplantation in the Prevention and Treatment of Fungal Infections guidelines of the Japan Society for Hematopoietic Cell Transplantation [14]. Due to their higher fungicidal activity compared to other antifungals, such as azoles [9] and fewer side effects and interactions [15], echinocandin antifungals, including micafungin, are frequently used in daily practice for those at a higher risk and severity of *Candida* infection after onset. Furthermore, micafungin is more frequently used than caspofungin in Japan [16]. Micafungin was marketed approximately 6 years earlier than caspofungin and there is an associated brand familiarity for this antifungal agent.

In a survey of detected *Candida* species in blood cultures, the percentage of detected *Candida* species in Japan was the highest for *C. albicans*, followed by *C. parapsilosis* and *C. glabrata* [17,18,19]. In the USA, *C. albicans* is also ranked first; however, this is followed by *C. glabrata* [20]. Forrest et al. studied the use of caspofungin and the frequency of detection of *C. parapsilosis* [21], and high micafungin use was considered one reason for the high frequency of detection of *C. parapsilosis* in Japan. To prove this hypothesis, it is necessary to examine whether similar results can be obtained in Japan, where micafungin use is high. No previous studies have investigated whether prior micafungin use is a risk factor for *C. parapsilosis* detection. In recent years, there are studies on the resistance of non-*parapsilosis Candida* species to echinocandin antifungals [22]. It is helpful to promote antifungal stewardship to clarify the current status of non-*parapsilosis Candida* species low susceptibility by prior administration of micafungin.

The present study investigated the causative species of candidemia and their drug susceptibility, and the use of antifungal agents. Risk factors associated with detecting *C. parapsilosis* and increasing the minimum inhibitory concentration (MIC) of non-*parapsilosis Candida* species were also investigated.

## 2. Materials and Methods

### 2.1. Ethics

This study was approved by the Yokohama City University Ethics Committee (approval number: B190600046, 8 August 2019).

### 2.2. Patients and Episode

Records from the microbiology laboratory were evaluated to identify patients with positive peripheral blood cultures (including central venous (CV) catheters) for *Candida* spp. from January 2010 to December 2018. Isolation of *Candida* spp. from at least one blood culture of a patient was defined as candidemia. If the same species was detected two times or more in the same patient, only the first time was included in the analysis.

Survey items included clinical outcomes at 30 days after positive cultures, patient characteristics, and drug prescription status. Patient background information included gender, previous hospitalization, stay in the intensive care unit (ICU), comorbidities, and history of surgery (within 90 days before candidemia onset) and drug exposure (administration for at least 2 days within 14 days before candidemia onset).

### 2.3. Organism Identification and Susceptibility Testing

Blood cultures were performed using the BacT/alert 3D system (bioMérieux, Lyon, France). All fungal isolates from blood cultures were identified with VITEK^TM^2 (bioMérieux, Lyon, France) using CHROMagar^TM^ *Candida* broth (Becton Dickinson Japan, Tokyo, Japan).

The MIC measurement followed the methodology of the Clinical and Laboratory Standards Institute (CLSI) M27-A3 and used yeast-like fungi DP-Eiken (Tokyo, Japan), with higher values adopted when MICs differed in the same isolate. The MIC measurement ranges were as follows: fluconazole 0.12–64 µg/mL, itraconazole and voriconazole 0.015–8 µg/mL, amphotericin B and caspofungin 0.03–16 µg/mL, and micafungin 0.015–16 µg/mL. Posaconazole and anidulafungin were not approved in Japan during the study period. MIC_50_ and MIC_90_ were calculated for each species. MIC_50_ and MIC_90_ are defined as the concentrations of each antifungal agent necessary to inhibit 50% and 90% of the isolates, respectively. MIC > 0.06 µg/mL was used as a criterion for the low susceptibility of micafungin for non-*parapsilosis Candida* species. MIC > 0.06 µg/mL was set with reference to the resistance norm of *C. glabrata* in CLSI M60 1st Edition (*Performance Standards for Antifungal Susceptibility Testing of Yeasts*) [23].

### 2.4. Antifungal Use

Clinical departments were divided into four categories: hematology, internal medicine, surgery, and others, and antifungal use was calculated by dividing the days of therapy by 1000 patient days (PDs) [24].

### 2.5. Factorial Analysis

Multivariate analyses were performed on isolates with available patient backgrounds to determine factors that increased the risk of *C. parapsilosis* detection and those that resulted in low susceptibility to micafungin (MIC > 0.06 µg/mL) among non-*parapsilosis Candida* species.

### 2.6. Statistical Analysis

Categorical variables were analyzed using Fisher’s exact tests. Logistic regression analysis was applied to identify demographic and clinical variables associated with *C. parapsilosis* and with candidemia with MIC of micafugnin > 0.06 μg/mL. Variables with a *p* < 0.20 by bivariate analysis were included in multivariable model selection. Model selection was conducted using stepwise logistic regression and consideration of 2-way interaction terms. The level of significance was set at α = 0.05. All statistical analyses were performed using the statistical software package IBM-SPSS statistics 26.0 (IBM, New York, NY, USA).

## 3. Results

### 3.1. Species Distribution of the Isolates and Mortality Rate

The overall species distribution is shown in Table 1. During the study period, a total of 153 *Candida* clinical blood isolates were collected. *C. albicans* was the most prevalent species, followed by *C. parapsilosis*, *C. glabrata*, *C. tropicalis,* and *C. famata*. These five species accounted for more than 90% of all isolates. Twelve isolates were unidentified to the species level. Thirty-day mortality was 23.5% overall and especially more than 30% in *C. tropicalis*, *C. famata*, and *C. krusei*.

### 3.2. Amount of Antifungal Usage and Species Distribution of Blood Isolated Candidemia from 2010 to 2018

Antifungal use in days of therapy (DOT)/1000 patient days (PDs) by department was 673.0, 15.2, 20.2, and 65.0 for hematology, internal medicine, surgery, and others, respectively (Table 2). When compared by the drug, the hematology department had the highest amount of antifungal use for azoles, echinocandins, and liposomal amphotericin B.

In terms of species distribution by department, *C. albicans* was lower (9%) and *C. parapsilosis* was higher (41%) in the hematology department than in the other departments (Figure 1). The numbers of isolates detected during the survey were 22, 42, 71, and 18 for hematology, internal medicine, surgery, and others, respectively. For the number of isolates per 10,000 patient days (PDs), hematology was the highest (3.0), followed by surgery (1.4), internal medicine (0.84), and others (0.3) (Figure 1). In hematology, the rate of prophylaxis was 95.5%.

### 3.3. Factorial Analysis for C. parapsilosis Associated with Candidemia

Univariate and multivariate analyses were performed on 147 isolates for factors that increased the risk of detecting *C. parapsilosis* (Table 3). Since detailed information was not available, six isolates were excluded in this analysis. Prior micafungin use increased the frequency of detecting *C. parapsilosis* and the multivariate analysis revealed that prior micafungin use was a risk factor for *C. parapsilosis* detection (odds ratio (OR) = 4.22; 95% confidence interval (CI) = 1.39–12.78; *p* = 0.011). ICU stay significantly decreased the frequency of detecting *C. parapsilosis* and multivariate analysis revealed that ICU stay was a risk factor for the onset of non-*parapsilosis Candida* species (OR = 0.276; 95% CI = 0.094–0.809; *p* = 0.019). Chronic renal disease also significantly decreased the frequency of detecting *C. parapsilosis*; however, multivariate analysis revealed no significant differences (OR = 0.441; 95% CI = 0.190–1.027; *p* = 0.058). Admission to the internal medicine department significantly decreased the frequency of detecting *C. parapsilosis*. However, patient department was not included in the multivariate analysis due to multicollinearity with micafungin exposure. There were no differences between the two groups regarding gender, history of hospitalization within 90 days, diabetes mellitus, organ transplantation, hematopoietic stem cell transplantation, neutropenia, gastrointestinal surgery, renal replacement therapy, and the administration of antimicrobials, steroids, and immunosuppressants.

### 3.4. Antifungal MIC Distribution of Candida Blood Isolates

The MIC_90_ of caspofungin for *Candida* spp. was 0.5–4 µg/mL and the MIC_90_ of micafungin was 0.03–1 µg/mL (Table 4). The MIC_50_ of caspofungin and micafungin for *C. parapsilosis* was 1 µg/mL and 0.5 µg/mL, respectively, and the MIC_90_ of both was 1 µg/mL. Apart from *C. parapsilosis*, *C. glabrata* showed higher MICs compared to the other species. The MIC_50_ of caspofungin and micafungin for *C. glabrata* was 1 µg/mL and 0.03 µg/mL, whereas the MIC_90_ was 4 µg/mL and 1 µg/mL, respectively.

### 3.5. Factorial Analysis for Low Micafungin Susceptibility in Non-parapsilosis Candida Species

Among non-*parapsilosis Candida* species, we performed univariate and multivariate analyses in 108 isolates to determine the factors leading to decreased susceptibility of micafungin (MIC > 0.06 µg/mL). In hematology (MIC ≤ 0.06 µg/mL: 5.0% vs MIC > 0.06 µg/mL: 32.1%; *p* < 0.001), neutropenia (MIC ≤ 0.06 µg/mL: 3.8% vs MIC > 0.06 µg/mL: 21.4%; *p* = 0.009), CV catheterization (MIC ≤ 0.06 µg/mL: 72.5% vs MIC > 0.06 µg/mL: 92.9%; *p* = 0.033), prior micafungin use (MIC ≤ 0.06 µg/mL: 3.8% vs MIC > 0.06 µg/mL: 35.7%; *p* < 0.001), and immunosuppressant use (MIC ≤ 0.06 µg/mL: 6.3% vs MIC > 0.06 µg/mL: 21.4%; *p* = 0.033) the detection frequency was significantly higher in the MIC > 0.06 µg/mL isolates (Table 5). Multivariate analysis revealed that prior micafungin use was a significant risk factor for an increased frequency of detecting isolates with MIC > 0.06 µg/mL (OR = 13.24; 95% CI = 3.23–54.2; *p* < 0.01). Patient department was not included in the multivariate analysis due to multicollinearity with micafungin exposure. There were no differences regarding gender, a history of hospital stay within 90 days, ICU stay, diabetes mellitus, organ transplantation, hematopoietic stem cell transplantation, gastrointestinal surgery, renal replacement therapy, CV port and administration of antibiotics or steroids between the two groups.

## 4. Discussion

The percentage of detected species (Table 1) was similar with previous studies from Japan [17,18,19], while it differed from the results from the United States. In the USA hospital-based antifungal use survey [25], echinocandin antifungal use accounted for 14% of all antifungals, whereas in the present study, echinocandin use was 25.1%. This difference in antifungal use affected the detected fungal species. The present study used mainly micafungin among echinocandins and the frequency of *C. parapsilosis* occurrence was elevated, which is similar with the study predominantly used caspofungin [21]. For the first time to date, we showed that increasing micafungin use increased the frequency of *C. parapsilosis* occurrence. While prior administration of fluconazole has been reported to be a risk factor for the breakthrough of *C. glabrata* and *C. krusei* [26], we showed for the first time that prior micafungin use is a risk factor for the breakthrough of *C. parapsilosis*.

In the antifungal use by department, the use of antifungal drugs was the highest in the hematology department. This department also had the highest detection frequency of non-*albicans Candida* species such as *C. parapsilosis*. Patients with hematological malignancy use more antifungal agents because antifungal prophylaxis is recommended in many guidelines [12,14,27,28,29]. Patients with hematological malignancy have a higher risk and frequency of developing deep mycoses. Fluconazole, itraconazole, voriconazole, and micafungin are recommended as prophylactic agents in the Japanese guidelines [12,14]. However, fluconazole is not active against *Aspergillus* spp. and itraconazole can often not be continued due to gastrointestinal toxicity [30] despite improved absorption with oral solutions. In CYP2C19 poor metabolizers, blood levels of voriconazole are likely to increase, with the percentage of poor metabolizers higher in Japanese than in Western individuals. Therefore, Japanese patients develop hepatic dysfunction more frequently due to voriconazole [31]. For these reasons, micafungin is often chosen for prophylaxis in Japan. Arendrup et al. reported an increase of non-*albicans Candida* species when the duration of antifungal use was at least 7 days prior to the detection of culture outcome [1,32]. In our study, prophylaxis was provided in 95.5% of hematology department isolates. The higher use of antifungals, including prophylaxis, was the reason for the higher rate of detecting non-*albicans Candida* species. In addition, the present study revealed that prior micafungin use is a risk factor for low susceptibility (Table 5). Compared with the previous study result, MIC_90_ of *C. glabrata* was 0.06–0.25 µg/mL in a Japanese study [17,19]; therefore, our results showed MIC_90_ for *C. glabrata* increased. The same as breakthrough infections with higher MICs occurring during micafungin use in *C. glabrata* [33], we showed prior micafungin use not only increased the frequency of *C. parapsilosis* but also is associated with low susceptibility in non-*parapsilosis Candida* species. The criteria for susceptibility remain controversial; therefore, in the present study, low susceptibility was defined as MIC > 0.06 μg/mL with reference to the criteria for resistance of *C. glabrata* in CLSI M60 1st Edition [23]. Andes et al. has reported an MIC of 0.06 µg/mL for ≥ 90% therapeutic efficacy at micafungin 100 mg/day administration considering the pharmacokinetics/pharmacodynamics (PK/PD) parameters [34]; as such, we considered it reasonable to define low susceptibility as MIC > 0.06 µg/mL.

Our results showed that the overall 30-day mortality rate was 23.5%, which was lower than other studies on overall mortality (39.3%) of nosocomial bloodstream infections caused by *Candida* in Japan [2]. Candidemia caused by *C. parapsilosis* is associated with a lower mortality rate [35,36], which could be attributed to the higher frequency of *C. parapsilosis* occurrence in our study. In contrast, the mortality rates of candidemia caused by *C. tropicalis*, *C. famata*, and *C. krusei* were as high as 36.4%, 50.0%, and 60.0%, respectively. Previous studies also showed that *C. tropicalis* and *C. krusei* candidemia had mortality rates of 43.1% and 58.7% [3], indicating similar results.

The present study had several limitations. First, it was a single-center retrospective study. A multicenter observational study [18] was conducted in Japan; however, it did not investigate whether antifungal agents affected the detected species or their susceptibilities. The results of our multivariate analyses might be influenced by the sample size and number of variables included in the models. In the future, similar considerations should be made in multiple centers with differing antifungal use status. Second, low susceptibility was defined as MIC > 0.06 µg/mL in the present study. The possibility of differing PK/PD parameters for each species has also been reported [37,38]; as such, MIC > 0.06 µg/mL is not the same as resistance. Third, fungal isolates were identified with only VITEKTM2 using the CHROMagarTM *Candida* broth. There are several reports of misidentification especially when *C. famata* is reported [39,40,41,42], so the possibility of misidentification must be considered. Although there are limitations, this is the first informative study to show that prior micafungin use affects the detected species and their respective MICs.

## 5. Conclusions

Prior micafungin use increased the risk of *C. parapsilosis* and the MIC > 0.06 µg/mL of micafungin in non-*parapsilosis Candida* species. Since there are only a few antifungal options for treatment, further antifungal stewardship considering azole-based antifungal use is required.

## Figures and Tables

**Figure 1 jof-07-00596-f001:**
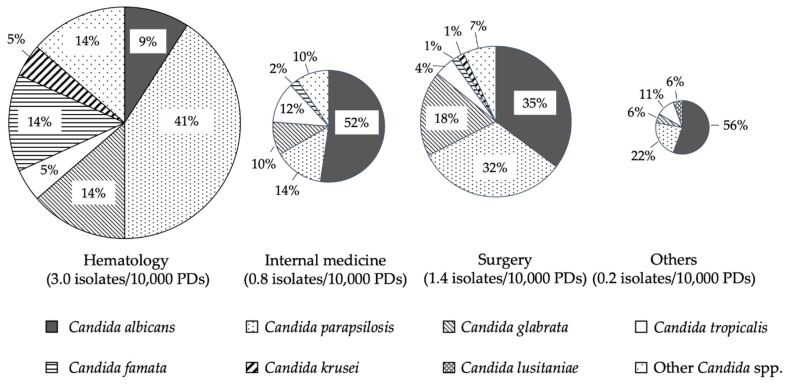
Species distribution of Candida blood isolates among different hospital departments from 2010 to 2018. Species distribution by clinical department is shown. The area ratio of the pie chart shows the number of cases per 10,000 patient days (PDs). The rate number is shown in the figure. Note: The total is not 100% because the data are rounded to integers.

**Table 1 jof-07-00596-t001:** Species distribution of *Candida* blood isolates.

Species	No. (%) of Isolates	30-Day Mortality Rates
*Candida albicans*	59 (38.6)	23.7
*Candida parapsilosis*	42 (27.5)	19.0
*Candida glabrata*	21 (13.7)	23.8
*Candida tropicalis*	11 (7.2)	36.4
*Candida famata*	5 (3.3)	60.0
*Candida krusei*	2 (1.3)	50.0
*Candida lusitaniae*	1 (0.7)	0
Other ^†^	12 (7.8)	8.3
Total	153 (100)	23.5

^†^ Not identified to species further than the genus *Candida*.

**Table 2 jof-07-00596-t002:** Antifungal use in days of therapy (DOT)/1000 patient days (PDs) among different hospital departments from 2010 to 2018.

	Hematology	Internal Medicine	Surgery	Others
Micafungin	181.9	4.3	10.7	9.2
Caspofungin	5.9	0.1	0.2	0.2
Fluconazole	104.1	5.1	7.1	42.7
Fosfluconazole	0.3	0.1	0.1	0.0
Itraconazole	332.6	2.0	0.3	4.1
Voriconazole	38.8	2.6	0.9	6.6
Liposomal	9.3	0.6	1.1	2.0
Amphotericin B				
Amphotericin B	0.0	0.0	0.0	0.1
Flucytosine	0.0	0.4	0.0	0.1
Total	673.0	15.2	20.2	65.0

The total does not account for patients who may have received multiple drugs on the same day.

**Table 3 jof-07-00596-t003:** Risk factors for bloodstream infections caused by *Candida parapsilosis*. (bivariate and multivariate analyses).

Factor	Non-*parapsilosis Candida* Species (*n* = 108)	*C. parapsilosis* (*n* = 39)	*p*	Unadjusted OR (95% CI)	*p*	Adjusted OR (95% CI)	*p*
No. (%) male patients	69 (64)	27 (69)	0.695				
Hospitalization in prior 90 days	56 (52)	16 (55)	0.267				
ICU stay	**38 (35)**	**6 (15)**	**0.025**	**0.307** **(0.102–0.923)**	**0.036**	**0.276** **(0.094–0.809)**	**0.019**
No. (%) of patient department:							
Hematology	13 (12)	7 (18)	0.415				
Internal medicine	**35 (32)**	**5 (13)**	**0.021**				
Surgery	46 (43)	23 (56)	0.094				
Others	14 (13)	4 (10)	0.781				
No. (%) of patients with:							
Diabetes	23 (21)	6 (15)	0.490				
Chronic renal disease	**54 (50)**	**12 (31)**	**0.041**	0.443 (0.187–1.05)	0.063	0.441 (0.190–1.027)	0.058
HIV infection	0 (0)	0 (0)	-				
Solid organ transplantation	2 (2)	0 (0)	1.00				
Bone marrow transplantation	4 (4)	4 (10)	0.209				
Neutropenia (< 500/μL)	9 (8)	4 (10)	0.746				
No. (%) of patients who underwent invasive procedures							
Gastrointestinal surgery	13 (12)	5 (13)	1.00				
Renal replacement therapy	17 (16)	3 (8)	0.281				
Tunneled catheter	3 (3)	4 (10)	0.081	2.17 (0.383–12.2)	0.382		
Nontunneled catheter	84 (78)	26 (67)	0.198	0.6 (0.228–1.58)	0.302		
No. (%) of patients with previous:							
Antibiotic treatment	78 (72)	24 (62)	0.229				
Fluconazole exposure	2 (2)	0 (0)	1.00				
Itraconazole exposure	2 (2)	1 (3)	1.00				
Voriconazole exposure	1 (1)	0 (0)	1.00				
Micafungin exposure	13 (12)	9 (23)	0.118	**4.97** **(1.56–15.5)**	**0.006**	**4.22** **(1.39–12.8)**	**0.011**
Liposomal Amphotericin B exposure	1 (1)	0 (0)	1.00				
Corticosteroid treatment	29 (27)	11 (28)	1.00				
Immunosuppression medications	11 (10)	6 (15)	0.391				

Abbreviations: CI, confidence interval; ICU, intensive care unit; HIV, human immunodeficiency virus. Bivariate analyses were analyzed using Fisher’s exact tests. Variables with a *p* < 0.20 by bivariate analysis were included in multivariable model selection. Model selection was conducted using stepwise logistic regression and consideration of 2-way interaction terms. The level of significance was set at α = 0.05.

**Table 4 jof-07-00596-t004:** Minimum inhibitory concentrations (MICs) of *Candida* blood isolates after 24 h of incubation.

Species and Antifungal Agent (No. of Isolates)	No. of Isolates with MIC (mg/mL) of:														MIC_50_ (μg/mL)	MIC_90_ (μg/mL)
	0.015	0.03	0.06	0.12	0.25	0.5	1	2	4	8	16	32	64	>64		
*C. albicans* (57)																
Fluconazole				19	18	6	8	3	1				1	1	0.25	2
Itraconazole	5	14	21	11	2	3					1 ^†^				0.06	0.25
Voriconazole	35	10	6	2	1	2					1 ^†^				≤ 0.01	0.12
Amphotericin B		1	1	4	24	22	4	1							0.25	0.5
Caspofungin ^‡^				2	12	3									0.25	0.5
Micafungin	49	7			1										≤ 0.01	0.03
*C. parapsilosis* (39)																
Fluconazole				1	5	20	12				1				0.5	1
Itraconazole		3	14	19	2	1									0.12	0.12
Voriconazole	23	15		1											0.015	0.03
Amphotericin B				6	13	18	2								0.5	0.5
Caspofungin ^‡^						2	7								1	1
Micafungin	1			1	4	23	10								0.5	1
*C. glabarata* (20)																
Fluconazole						1	4		7	6	2				4	16
Itraconazole			1	3	6	6	4								0.25	1
Voriconazole	3	1	4	5	6	1									0.12	0.25
Amphotericin B				1	5	11	3								0.5	1
Caspofungin ^‡^						3	3		1						1	4
Micafungin	8	3	1	1	2	2	3								0.03	1
*C. tropicalis* (11)																
Fluconazole					4	2	1	1					1	2	0.5	> 64
Itraconazole		1	5	1	1	1		1			1 ^†^				0.06	2
Voriconazole	2	4		1	1		1		1		1 ^†^				0.03	4
Amphotericin B					3	8									0.5	0.5
Caspofungin ^‡^					2	5									0.5	0.5
Micafungin	4	7													0.03	0.03
*C. famata* (5)																
Fluconazole				1			1	3							2	2
Itraconazole			1		4										0.25	0.25
Voriconazole	1	1	3												0.06	0.06
Amphotericin B					3	2									0.25	0.5
Caspofungin ^‡^					2	1	2								0.5	1
Micafungin				2	2	1									0.25	0.5
*C. krusei* (2)																
Fluconazole ^§^											1	1			16	32
Itraconazole					2										0.25	0.25
Voriconazole					2										0.25	0.25
Amphotericin B						1	1								0.5	1
Caspofungin ^‡^							2								1	1
Micafungin				2											0.12	0.12
Other *Candida* spp. (13)																
Fluconazole					3	4	1	3	2						0.5	4
Itraconazole			4	4	4	1									0.12	0.25
Voriconazole	6	1	4	1	1										0.03	0.12
Amphotericin B				1	8	3	1								0.25	0.5
Caspofungin ^‡^						3	5								1	1
Micafungin		1		3	3	4	2								0.25	1

^†^ The MIC values are > 8 mg/mL; ^‡^ susceptibility test of caspofungin was conducted in only 55 cases; ^§^ *C. krusei* is intrinsically resistant to fluconazole. The gray color indicates outside of MIC measurement ranges.

**Table 5 jof-07-00596-t005:** Risk factors for bloodstream infections caused by non-*parapsilosis Candida* species with a minimum inhibitory concentration (MIC) of micafungin >0.06 μg/mL. (Bivariate and multivariate analyses.)

Factor	MIC ≤ 0.06 μg/mL (*n* = 80)	MIC > 0.06 μg/mL (*n* = 28)	*p*	Unadjusted OR (95% CI)	*p*	Adjusted OR (95% CI)	*p*
No. (%) male patients	51 (64)	18 (64)	1.00				
Hospitalization in prior 90 days	38 (48)	16 (57)	0.511				
ICU stay	26 (33)	12 (43)	0.362				
No. (%) of patient department:							
Hematology	**4 (5)**	**9 (32)**	**< 0.001**				
Internal medicine	30 (38)	5 (18)	0.064				
Surgery	33 (41)	13 (46)	0.662				
Others	13 (16)	1 (4)	0.109				
No. (%) of patients with:							
Diabetes	15 (19)	8 (29)	0.292				
Chronic renal disease	39 (49)	15 (54)	0.669				
HIV infection	0 (0)	0 (0)	-				
Solid organ transplantation	1 (1)	1 (4)	0.453				
Bone marrow transplantation	2 (3)	2 (7)	0.276				
Neutropenia(< 500/μL)	**3 (4)**	**6 (21)**	**0.009**	0.762 (0.094–6.18)	0.799		
No. (%) of patients who underwent invasive procedures							
Gastrointestinal surgery	11 (14)	2 (7)	0.508				
Renal replacement therapy	12 (15)	5 (18)	0.766				
Tunneled catheter	3 (4)	0 (0)	0.567				
Nontunneled catheter	**58 (73)**	**26 (93)**	**0.033**	4.61 (0.846–25.1)	0.077		
No. (%) of patients with previous:							
Antibiotic treatment	55 (69)	23 (82)	0.224				
Micafungin exposure	**3 (4)**	**10 (36)**	**< 0.001**	**11.5** **(1.91–69.1)**	**0.008**	**13.2** **(3.23–54.2)**	**< 0.01**
Corticosteroid treatment	20 (25)	9 (32)	0.467				
Immunosuppression medications	**5 (6)**	**6 (21)**	**0.032**	**5.44** **(1.05–28.1)**	**0.043**	3.44 (0.831–14.2)	0.088

Abbreviations: CI, confidence interval; ICU, intensive care unit; HIV, human immunodeficiency Virus. Bivariate analyses were analyzed using Fisher’s exact tests. Variables with a *p* < 0.20 by bivariate analysis were included in multivariable model selection. Model selection was conducted using stepwise logistic regression and consideration of 2-way interaction terms. The level of significance was set at α = 0.05.

## Data Availability

The data presented in this study are available on request from the corresponding author. The data are not publicly available due to privacy.

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
