# Peer review of "Species Distribution of Candidemia and Their Susceptibility in a Single Japanese University Hospital: Prior Micafungin Use Affects the Appearance of Candida parapsilosis and Elevation of Micafungin MICs in Non-parapsilosis Candida Species"

_jof, 2021, doi:10.3390/jof7080596_

Round 1
Reviewer 1 Report
Sakamoto, et al. present analysis of candidemia isolates from a Japanese University Hospital, with particular focus on the risk factors for C. parapsilosis as well as for elevated micafungin MICs in non-C. parapsilosis isolates. This well-written manuscript includes rich data analysis, including detailed tables and figures, presenting a great insight into species and MIC distribution, 30-day mortality by species, and differences in antifungal use by department. As a reader, clinical department felt like ‘the elephant in the room’ in many of the analyses, though, which could perhaps be resolved with more incorporation/exploration of clinical department or with an explanation of the decisions the authors made regarding confounders. There may also be room for clarification on the significance of choices surrounding the C. parapsilosis v. non-parapsilosis Candida species divide. Please find some considerations for the authors below:
Keywords
- Line 42: Misspelled non-albicans
Introduction
- Line 48: “…with a higher mortality rate than other infections in the world” sounds quite vague. Is this referring to other bloodstream nosocomial infections or any other infection?
- Line 69-71: This sentence says bacterial species, but is referring to different fungal/yeast/Candida species instead. “Bacterial” should not be used to describe fungi.
Methods
- Line 112-115: It would help to state here that this is only for non-parapsilosis Candida since CLSI M60 sets different micafungin resistance cut-offs for different species. While C. glabrata has MIC <= 0.06 as susceptible, 0.12 as intermediate and >= 0.25 as resistant, C. parapsilosis, which is the main focus of this paper, is set at <=2 as susceptible, 4 as intermediate, and >=8 as resistant. Therefore, calling anything >0.06 for C. parapsilosis would be problematic and not adherent to accepted MIC interpretations, so it would be best to avoid such confusion by the readers.
- Line 97-98: What was the timing drug exposure? And was there any indication of whether the drug was given for prophylaxis or empiric treatment or treatment post-identification of an infection? It sounds like this might be available (based on line 241-242) but is not presented in results.
Results
- Line 138: C. famata is commonly associated with misidentifications using VITEK 2, so it may be good to include a caveat about C. famata being in the top species found. Here is a quick list of a few articles discussing this issue:
- https://pubmed.ncbi.nlm.nih.gov/24977144/
- https://pubmed.ncbi.nlm.nih.gov/23100350/
- https://pubmed.ncbi.nlm.nih.gov/33250336/
- https://pubmed.ncbi.nlm.nih.gov/20944497/
- Table 2: It appears that the total was not considering that patients may have had combination therapy, having overlapping drug courses. One could see how expansion of this logic could mean that days of therapy could exceed the 1000 patient days, though this did not actually happen in the data. Because of this, it may be prudent to either remove the total row, recalculate the total accounting for patients on multiple drugs at once, or at least add a footnote clarifying that the total does not account for patients who may have had multiple drugs at one time.
- Section 3.3: It might be easier for readers if the authors adhered to describing the increases or decreases in risk for C. parapsilosis, rather than flipping back and forth between framing the findings as risk for C. parapsilosis v. risk for non-parapsilosis Candida.
- Table 3: Was clinical department considered for the analysis at all? Based Table 2, it seems like there is a very close relationship between antifungal use and clinical department. Some of the clinical factors within Table 3 could possibly serve as partial proxies for clinical department, but are not quite the same. Given the relationships presented in Table 2 and Figure 1 and because C. parapsilosis can be transmitted (though most often reported in neonatal ICUs) meaning proximity or epidemiologic links may play a role, I am quite interested in learning how the clinical department might play into this, or at least an explanation of why that was not considered or presented.
- Section 3.4: It is not clear why the detail focus here is on C. glabrata when the manuscript’s focus is on C. parapsilosis instead.
- Table 4: C. krusei is intrinsically resistant to fluconazole, so it may be best not to present the MICs for that combination, or else, to present it with a footnote.
- Table 4: Given that the species is needed for proper interpretation of MICs, it does not seem like there is much value added in presenting the ‘all’ section on this table.
- Table 5: I would recommend rephrasing the title to “Risk factors for bloodstream infections caused by non-parapsilosis Candida species with minimum inhibitory concentration (MIC) of micafungin >0.06 μg/mL” so that it is clear that the species, not the MIC caused the infection.
- Table 5: Misspelled adjusted
- Table 5: As with table 3, it seems like much of this, including prior micafungin exposure, would be tied to clinical department, especially based on the micafungin guidelines presented in the introduction, so it might be interesting to see data on clinical department or present the explanation of why that was not done for the reader.
- Overall: It is not particularly clear to me why C. parapsilosis isolates were removed from the non-susceptibility analyses, especially given that echinocandin resistance in C. parapsilosis is so rare. Is it because the MICs for C. parapsilosis typically follow a different pattern than other Candida species? It seems like the same analysis could have been conducted, including C. parapsilosis, using the CLSI breakpoints for those species with breakpoints and for MIC <=/> 0.06 for those without established breakpoints.
Discussion
- Line 244-245: This statement feels too strong, as it is known that patients can develop echinocandin resistance (or elevated echinocandin MICs) rapidly on treatment.
Author Response
Sakamoto, et al. present analysis of candidemia isolates from a Japanese University Hospital, with particular focus on the risk factors for C. parapsilosis as well as for elevated micafungin MICs in non-C. parapsilosis isolates. This well-written manuscript includes rich data analysis, including detailed tables and figures, presenting a great insight into species and MIC distribution, 30-day mortality by species, and differences in antifungal use by department. As a reader, clinical department felt like ‘the elephant in the room’ in many of the analyses, though, which could perhaps be resolved with more incorporation/exploration of clinical department or with an explanation of the decisions the authors made regarding confounders. There may also be room for clarification on the significance of choices surrounding the C. parapsilosis v. non-parapsilosis Candida species divide. Please find some considerations for the authors below:
Keywords
Line 42: Misspelled non-albicans
Reply:
We apologize for the mistake. We revised the main text accordingly. (line 42)
Introduction
Line 48: “…with a higher mortality rate than other infections in the world” sounds quite vague. Is this referring to other bloodstream nosocomial infections or any other infection?
Reply:
We appreciate your comments. Per your suggestion, we revised the main text as follows.
‘Candidemia is one cause of nosocomial bloodstream nosocomial infections [1]. The mortality rate from candidemia was 39.3% in Japan [2], 38%–39.2% in the United States [3-4], and 19%–38.8% in Europe [5-6] with a higher mortality rate than other nosocomial bloodstream infections in the world.’ (line 46-49)
Line 69-71: This sentence says bacterial species, but is referring to different fungal/yeast/Candida species instead. “Bacterial” should not be used to describe fungi.
Reply:
We thank you for your suggestion. As per your suggestion, we revised our main text as follows.
‘In a survey of detected bacterial Candida species of Candida in blood cultures, the percentage of detected Candida bacterial species in Japan was the highest for C. albicans, followed by C. parapsilosis and C. glabrata [17-19].’ (line 69)
Methods
Line 112-115: It would help to state here that this is only for non-parapsilosis Candida since CLSI M60 sets different micafungin resistance cut-offs for different species. While C. glabrata has MIC <= 0.06 as susceptible, 0.12 as intermediate and >= 0.25 as resistant, C. parapsilosis, which is the main focus of this paper, is set at <=2 as susceptible, 4 as intermediate, and >=8 as resistant. Therefore, calling anything >0.06 for C. parapsilosis would be problematic and not adherent to accepted MIC interpretations, so it would be best to avoid such confusion by the readers.
Reply:
We thank you for your suggestion. As per your suggestion, we revised our main text as follows.
‘MIC > 0.06 µg/mL was used as a criterion for low susceptibility of micafungin for non-parapsilosis Candida species.’ (line 114)
Line 97-98: What was the timing drug exposure? And was there any indication of whether the drug was given for prophylaxis or empiric treatment or treatment post-identification of an infection? It sounds like this might be available (based on line 241-242) but is not presented in results.
Reply:
We revised our main text to specify the timing of drug exposure as follows.
‘Patient background information included gender, previous hospitalization, stay in the intensive care unit (ICU), comorbidities, and history of surgery (within 90 days before candidemia onset) and drug exposure (administration for at least 2 days within 14 days before candidemia onset).’ (line 98-99)
In addition, the rate of prophylaxis was calculated to assess the risk of species distribution, and the main text was revised as follows.
‘In hematology, the rate of prophylaxis was 95.5%.’ (line 156-157)
Results
Line 138: C. famata is commonly associated with misidentifications using VITEK 2, so it may be good to include a caveat about C. famata being in the top species found. Here is a quick list of a few articles discussing this issue:
https://pubmed.ncbi.nlm.nih.gov/24977144/
https://pubmed.ncbi.nlm.nih.gov/23100350/
https://pubmed.ncbi.nlm.nih.gov/33250336/
https://pubmed.ncbi.nlm.nih.gov/20944497/
Reply:
We thank you for these insights. As per your suggestion, we added the following sentences to our main text.
‘Third, fungal isolates were identified with only VITEKTM2 using CHROMagarTM Candida broth. There are several reports of misidentification especially when C. famata is reported [39-42], so the possibility of misidentification must be considered.’ (line 283-285)
Table 2: It appears that the total was not considering that patients may have had combination therapy, having overlapping drug courses. One could see how expansion of this logic could mean that days of therapy could exceed the 1000 patient days, though this did not actually happen in the data. Because of this, it may be prudent to either remove the total row, recalculate the total accounting for patients on multiple drugs at once, or at least add a footnote clarifying that the total does not account for patients who may have had multiple drugs at one time.
Reply:
We thank you for your valuable comments. As you pointed out, it is assumed that there was overlap in combination therapy and treatment course. To clarify this, we would need to re-analyze all patient data, but we don’t have enough time before the revision is due (in 5 days). The total was calculated because it was thought to be necessary to discuss whether the amount of antifungal drug use was high or low in each department. The Japanese guideline recommends combination therapy for severe Aspergillus infection in children and Cryptococcus meningitis, but the frequency of this therapy is very low. In addition, the IDSA guidelines do not recommend combination therapy. Therefore, following your suggestion, we added the following sentence to the revised manuscript.
‘Total does not account for patients who may have received multiple drugs on the same day.’ (Table 2, footnote)
Section 3.3: It might be easier for readers if the authors adhered to describing the increases or decreases in risk for C. parapsilosis, rather than flipping back and forth between framing the findings as risk for C. parapsilosis v. risk for non-parapsilosis Candida.
Reply:
We thank you for your suggestion. As per your suggestion, we revised our main text as follows.
‘ICU stay significantly decreased increased the frequency of detecting C.non-parapsilosis Candida parapsilosis species, and multivariate analysis revealed that ICU stay was a risk factor for the onset of non-parapsilosis Candida species (OR 0.276; 95% CI 0.094 – 0.809; p = 0.019). Chronic renal disease also significantly decreased increased the frequency of detecting C. parapsilosis non-parapsilosis Candida species; however, multivariate analysis revealed no significant differences (OR 0.441; 95% CI 0.190 – 1.027; p = 0.058).’ (line 172-177)
Table 3: Was clinical department considered for the analysis at all? Based Table 2, it seems like there is a very close relationship between antifungal use and clinical department. Some of the clinical factors within Table 3 could possibly serve as partial proxies for clinical department, but are not quite the same. Given the relationships presented in Table 2 and Figure 1 and because C. parapsilosis can be transmitted (though most often reported in neonatal ICUs) meaning proximity or epidemiologic links may play a role, I am quite interested in learning how the clinical department might play into this, or at least an explanation of why that was not considered or presented.
Reply:
We thank you for your suggestion. As per your suggestion, we have revised the Table 3 and the main text (line 178-180). Departments were not included in the multivariate analysis due to multicollinearity with micafungin exposure.
‘Admission to the internal medicine department significantly decreased the frequency of detecting C. parapsilosis. However, patient department was not included in the multivariate analysis due to multicollinearity with micafungin exposure.’ (line 178-180)
Section 3.4: It is not clear why the detail focus here is on C. glabrata when the manuscript’s focus is on C. parapsilosis instead.
Reply:
We thank you for your suggestion. As per your suggestion, we added the following sentence to the revised manuscript.
‘The MIC50 of caspofungin and micafungin for C. parapsilosis was 1 µg/mL and 0.5 µg/mL, respectively, and the MIC90 of both was 1 µg/mL.’ (line 187-188)
Table 4: C. krusei is intrinsically resistant to fluconazole, so it may be best not to present the MICs for that combination, or else, to present it with a footnote.
Reply:
We thank you for your suggestion. As per your suggestion, we have revised Table 4 and the footnote.
Table 4: Given that the species is needed for proper interpretation of MICs, it does not seem like there is much value added in presenting the ‘all’ section on this table.
Reply:
We thank you for your suggestion. As you pointed out, the ‘all’ section is not necessary. However, the frequency of isolating Candida spp. at various MICs is considered to be helpful for empiric therapy prior to fungal identification. This was discussed previous reports [Takakura S. et al., J Antimicrob Chemother. 2004;53:283-9, Kakeya H. et al., Med Mycol J. 2018;59:E19-E22]. Therefore, we moved the ‘all’ section in supplementary Table S1.
Table 5: I would recommend rephrasing the title to “Risk factors for bloodstream infections caused by non-parapsilosis Candida species with minimum inhibitory concentration (MIC) of micafungin >0.06 μg/mL” so that it is clear that the species, not the MIC caused the infection.
Reply:
We thank you for your suggestion. As per your suggestion, we have revised the title and contents of Table 5.
Table 5: Misspelled adjusted
Reply:
We apologize for the mistake. We have revised the Table 3 and Table 5 accordingly.
Table 5: As with table 3, it seems like much of this, including prior micafungin exposure, would be tied to clinical department, especially based on the micafungin guidelines presented in the introduction, so it might be interesting to see data on clinical department or present the explanation of why that was not done for the reader.
Reply:
We thank you for your suggestion. Just as before, regarding your comment for Table 3, we have revised Table 5 and the main text (line 204-206, 213-214).Departments were not included in the multivariate analysis due to multicollinearity with micafungin exposure.
‘In hematology (MIC ≤ 0.06 µg/mL: 5.0% vs MIC > 0.06 µg/mL: 32.1%; p < 0.001), neutropenia (MIC ≤ 0.06 µg/mL: 3.8% vs MIC > 0.06 µg/mL: 21.4%; p = 0.009), CV catheterization ・・・.’ (line 205-206)
‘Patient department was not included in the multivariate analysis due to multicollinearity with micafungin exposure.’ (line 213-214)
Overall: It is not particularly clear to me why C. parapsilosis isolates were removed from the non-susceptibility analyses, especially given that echinocandin resistance in C. parapsilosis is so rare. Is it because the MICs for C. parapsilosis typically follow a different pattern than other Candida species? It seems like the same analysis could have been conducted, including C. parapsilosis, using the CLSI breakpoints for those species with breakpoints and for MIC <=/> 0.06 for those without established breakpoints.
Reply:
We thank you for valuable suggestion. As you pointed out, it is important to analyze all Candida spp. including C. parapsilosis. Other studies have conducted an analysis using the CLSI breakpoints [Garnacho-Montero, J. et al., Antimicrob. Agents Chemother. 2010, 54, 3149 - 3154]. Although different PK/PD parameters have been reported for different species of Candida spp., they have not been widely accepted in clinical practice and are controversial. Although susceptibility tests have been performed on non-protein-bound forms, micafungin has a very high protein binding rate of 99%. Therefore, the PK/PD theory indicated a MIC of 0.06 μ g/mL, which was used as a standard, and was applied in a package. However, in our results, 38 out of 39 C. parapsilosis isolates had a microfungin MIC > 0.06, so it was not possible to identify risk factors associated with low susceptibility of C. parapsilosis. Therefore, analysis of all Candida spp. including C. parapsilosis, is influenced by the factors of C. parapsilosis. In particular, micafungin exposure is a risk factor for C. parapsilosis, resulting in multicollinearity. Therefore, we performed low susceptibility factor analysis only for non-parapsilosis spp.
Discussion
Line 244-245: This statement feels too strong, as it is known that patients can develop echinocandin resistance (or elevated echinocandin MICs) rapidly on treatment.
Reply:
We thank you for your comments. As per your suggestion, we have revised the main text as follows.
“In addition, the present study revealed for the first time that prior micafungin use is a risk factor for low susceptibility (Table 6).” (line 254)

Reviewer 2 Report
The manuscript entitled: “Species distribution of candidemia and their susceptibility in a 2 single Japanese university hospital: prior micafungin use af-3 fects the appearance of Candida parapsilosis and elevation of 4 micafungin MICs in non-parapsilosis Candida species” by Yasutaka Sakamoto et al., is a study concerning the effect of micafungin use on changes in the detected Candida species in a single Japanese university hospital.
It is a retrospective study where the authors concluded that prior micafungin use increased the risk of C. parapsilosis and the MIC > 0.06 μg/mL of micafungin in non-parapsilosis Candida species. It is a valuable manuscript that it should be published in the Journal of Fungi
Author Response
The manuscript entitled: “Species distribution of candidemia and their susceptibility in a 2 single Japanese university hospital: prior micafungin use af-3 fects the appearance of Candida parapsilosis and elevation of 4 micafungin MICs in non-parapsilosis Candida species” by Yasutaka Sakamoto et al., is a study concerning the effect of micafungin use on changes in the detected Candida species in a single Japanese university hospital.
It is a retrospective study where the authors concluded that prior micafungin use increased the risk of C. parapsilosis and the MIC > 0.06 μg/mL of micafungin in non-parapsilosis Candida species. It is a valuable manuscript that it should be published in the Journal of Fungi
Reply:
We thank you for review of our manuscript. We are thankful for the time and energy you expended.

Reviewer 3 Report
This manuscript is well written and clearly addresses the topic of interest that the authors are aiming for. One concern is that the authors state on line 69 of the introduction," In a survey of detected bacterial species of Candida in blood cultures, the percentage of detected bacterial species in Japan...". Candida species are not bacterial but are fungal. Other than this statement I found the manuscript enjoyable to read, largely devoid of errors with only very very minor grammatical errors that a simple double check of the manuscript should be able to resolve.
Author Response
This manuscript is well written and clearly addresses the topic of interest that the authors are aiming for. One concern is that the authors state on line 69 of the introduction," In a survey of detected bacterial species of Candida in blood cultures, the percentage of detected bacterial species in Japan...". Candida species are not bacterial but are fungal. Other than this statement I found the manuscript enjoyable to read, largely devoid of errors with only very very minor grammatical errors that a simple double check of the manuscript should be able to resolve.
Reply:
We thank you for your suggestion. As per your suggestion, we revised our main text as follows.
“In a survey of detected bacterial Candida species of Candida in blood cultures, the percentage of detected Candida bacterial species in Japan was the highest for C. albicans, followed by C. parapsilosis and C. glabrata [17-19].” (line 69-70)
